# Linking Adiposity to Interstitial Lung Disease: The Role of the Dysfunctional Adipocyte and Inflammation

**DOI:** 10.3390/cells12182206

**Published:** 2023-09-05

**Authors:** Michael Macklin, Chelsea Thompson, Leticia Kawano-Dourado, Iazsmin Bauer Ventura, Camila Weschenfelder, Andrés Trostchansky, Aline Marcadenti, Robert M. Tighe

**Affiliations:** 1Section of Rheumatology, The University of Chicago, Chicago, IL 60637, USA; iazsmin.ventura@bsd.uchicago.edu; 2Hcor Research Institute (IP-Hcor), Hcor, São Paulo 04004-050, Brazil; ldourado@hcor.com.br (L.K.-D.); amarcaden@hcor.com.br (A.M.); 3Pulmonary Division, Heart Institute (InCor), University of Sao Paulo Medical School, São Paulo 05403-903, Brazil; 4Graduate Program in Health Sciences (Cardiology), Cardiology Institute, University Foundation of Cardiology (IC/FUC), Porto Alegre 90050-170, Brazil; camilawesche@gmail.com; 5Department of Biochemistry and Biomedical Research Center, School of Medicine, University of the Republic, Montevideo 11800, Uruguay; atrostchansky@gmail.com; 6Graduate Program in Epidemiology, School of Public Health, University of São Paulo (FSP-USP), São Paulo 01246-904, Brazil; 7Division of Pulmonary, Allergy, and Critical Care Medicine, Duke University Medical Center, Durham, NC 27710, USA; robert.tighe@duke.edu

**Keywords:** obesity, adiposity, interstitial lung disease, autoimmunity, fibrosis, inflammation

## Abstract

Adipose tissue has functions beyond its principal functions in energy storage, including endocrine and immune functions. When faced with a surplus of energy, the functions of adipose tissue expand by mechanisms that can be both adaptive and detrimental. These detrimental adipose tissue functions can alter normal hormonal signaling and promote local and systemic inflammation with wide-ranging consequences. Although the mechanisms by which adipose tissue triggers metabolic dysfunction and local inflammation have been well described, little is known about the relationship between adiposity and the pathogenesis of chronic lung conditions, such as interstitial lung disease (ILD). In this review, we detail the conditions and mechanisms by which adipose tissue becomes dysfunctional and relate this dysfunction to inflammatory changes observed in various forms of ILD. Finally, we review the existing basic and clinical science literature linking adiposity to ILD, highlighting the need for additional research on the mechanisms of adipocyte-mediated inflammation in ILD and its clinical implications.

## 1. Introduction

Interstitial lung diseases (ILDs) are a collection of chronic lung disorders defined by expansion of the distal lung parenchyma interstitium with inflammatory cells and/or fibrosis [1]. In many cases, ILD progresses to pulmonary fibrosis and is a source of significant morbidity and mortality [2]. ILDs include several different types, reflecting distinct etiologies. A large group of ILDs result from autoimmune diseases. Autoimmune-related ILDs refer to a broad group of interstitial pulmonary disorders caused by an underlying systemic autoimmune disease, such as rheumatoid arthritis, systemic sclerosis, and systemic lupus erythematosus [2]. These disorders are typically defined by evidence of lung inflammation thought to be due to persistent immune activation. Frequently, this immune system activation is directly attributed to systemic autoimmunity, but other factors likely augment this lung inflammation and worsen autoimmune-related ILDs. One such potential factor augmenting ILD severity is increased adiposity in ILD both associated with and not associated with systemic autoimmunity [3]. Although the molecular mechanisms involved in the pathogenesis of ILD are poorly understood, alterations in adipokines have been identified as possible mediators of the pro-inflammatory changes that lead to interstitial lung abnormalities and fibrosis [3,4]. Additionally, the cascade of dysfunction mediated by abnormal adipocyte tissue shares common cellular mechanisms with previously identified pathways of inflammation in ILD [3,4].

Excess adipocyte tissue mass, which defines adiposity, is referred to as “obesity” in popular culture and in epidemiological and medical research, and it is a major public health concern, representing an annual estimated cost of 260 billion dollars in the United States [5]. A significant challenge with obesity research is that the typical measure used to define obesity, the body mass index (BMI), does not reflect actual adipocyte tissue mass and distribution [5]. This fact is critical, as unhealthy alterations of visceral and subcutaneous adipose tissue deposits, not BMI, mediate the pro-inflammatory and deleterious effects of obesity [6,7]. A simple comparison of body weight relative to height is an imperfect surrogate to describe these unhealthy changes. Therefore, consideration of adiposity over BMI might provide better understanding of disease-mediated effects. Further supporting this assertion, research has shown that abnormal adipose tissue distribution contributes to metabolic dysfunction through multiple mechanisms, including the secretion of pro-inflammatory cytokines and endothelial activation triggered by free fatty acid absorption [8].

Although the specific relationships between excess abnormal adipose tissue and the development and severity of ILDs are limited, the aim of this review is to summarize potential common pathways of adipocyte tissue dysfunction and ILD, as well as report the literature describing the role of dysfunctional adipocytes in the pathogenesis and prognosis of ILD. A limitation of this review is that not every subtype of ILD is represented in the literature, with the vast majority of the information on this topic most relevant to idiopathic pulmonary fibrosis (IPF) and autoimmune-related ILD cases. The influence of adiposity has also not been well described in every autoimmune cause of ILD to date. Given this point, adiposity may not influence every distinct subtype of ILD, specifically hypersensitivity pneumonitis and other causes associated with particular occupational or environmental exposures, given their unique pathophysiology, and may not encompass every cause of autoimmune-related ILD. Finally, we highlight the gaps in the scientific literature to accurately understand the direct impact of the dysfunctional adipocyte in the development and progression of ILD.

## 2. Adiposity and the Dysfunctional Adipocyte

Adipose tissue is composed of a complex network of functionally diverse cell types, including adipocytes, connective tissue matrix, nerve tissue, stromal-vascular cells, and immune cells [9]. In addition to energy storage and release, adipose tissue acts as an endocrine organ, secreting bioactive circulating peptides that have a variety of metabolic effects on a diverse array of organs [10]. Adipose tissue is divided into white adipose tissue and brown adipose tissue. Brown adipose tissue contains abundant mitochondria and functions to facilitate adaptive thermogenesis [11,12]. White adipose tissue contains large adipocytes and fewer mitochondria, giving it the capacity for energy storage and homeostasis [13,14]. White adipose tissue functions as an endocrine organ, releasing hormones and responding to hormonal signals that modulate whole-body metabolism and insulin resistance [15,16].

Brown adipose tissue is found only in limited locations and quantities in adults. Subcutaneous locations include under the clavicles, in the axillary region, in the inguinal fossa, in the anterior abdominal wall, and in the neck [17]. Visceral areas include perivascular locations around solid organs and the trachea, esophagus, greater omentum, and transverse colon [17]. White adipose tissue can also be induced to form beige adipocytes from cold temperatures or β3 signaling [18]. These beige adipocytes have similar thermogenic qualities to brown adipose tissue [18]. Brown adipocytes, and by extension beige adipocytes, are thought to be protective against obesity and its resulting metabolic derangements by inducing energy utilization for thermogenesis [18]. It is an active area of study whether transplantation of brown adipose tissue or inducing beige adipose tissue may be helpful for treating obesity [18]. Given that the pathophysiology of white adipose tissue has been much more studied and well described, we focus in this review on the effects of white adipose tissue only unless otherwise specified. There is currently no clinical or preclinical data to suggest a relationship between lung inflammation and brown adipose tissue.

In a state of positive energy balance, white adipose tissue adipocytes undergo hyperplasia or hypertrophy to store excess energy. Hyperplasia, a process called adipogenesis, is a two-step process whereby mesenchymal stem cells restricted to the adipocyte lineage activate signaling pathways to form functional, insulin-responsive adipocytes [14]. Adipogenesis generates smaller, more numerous adipocytes that have the advantage of maintaining proper vascularity and limiting a pro-inflammatory and pro-oxidant milieu in adipose tissue. Alternatively, in adipocyte hypertrophy, adipose tissue expands outwardly to accommodate energy storage [19]. However, these adipocytes undergoing hypertrophy are more likely to reach a critical size threshold whereby angiogenesis is compromised, leading to adipose tissue inflammation and dysfunction [14]. Therefore, the balance between adipogenesis and adipocyte hypertrophy can influence the systemic effects of adiposity.

In white adipose tissue, the balance between adipogenesis or hypertrophy depends on whether expansion occurs in the visceral and subcutaneous body spaces. Excess subcutaneous adipose tissue is preferentially formed through adipogenesis. However, once the subcutaneous adipose tissue reaches a critical expansion threshold, visceral expansion starts to occur [20,21]. Visceral adipose tissue, which is mostly found in the intra-abdominal cavity, has a lower capacity for adipogenesis but rather favors expansion through hypertrophy [22]. These larger adipocytes are more metabolically active and prone to insulin resistance and inflammation [22]. Although increases in visceral and subcutaneous adipose tissue mass increase BMI, visceral adipose tissue mass expansion is more likely to worsen metabolic dysfunction and promote local inflammation and insulin resistance [23,24]. This fact highlights the importance of considering the distribution of adiposity, as increased visceral adipose tissue has a greater association with detrimental organ complications related to adiposity.

Expansion of visceral adipose tissue via hypertrophy alters its endocrine functions. Adipokines, which are adipocyte-produced bioactive factors with hormonal and regulatory immune functions, are key mediators of these dysfunctional changes. Alterations of adipokine ratios, specifically an excess of leptin and a paucity of adiponectin, promote insulin resistance and increased pro-inflammatory immune activity [25]. For example, exogenous replenishment of adiponectin improves glucose tolerance independent of body weight in obese mice [25]. Alternatively, leptin exerts pro-inflammatory effects by inducing monocyte expansion and cytokine secretion [26]. Thus, the adiponectin/leptin ratio has emerged as a biomarker of adipose tissue dysfunction [20]. Beyond leptin, excess adipose tissue mass also favors the production of other adipokines that promote insulin resistance and local inflammation [27]. These adipokines include resistin, progranulin, retinol-binding protein 4 (RBP4), fetuin-A, and lipocalin 2. Overall, the altered expression of adipokine ratios from the expansion of visceral adipose tissue likely are important contributors to organ dysfunction and the progression of metabolic complications from adiposity.

In addition to altered adipokine secretion, the development of oxidative stress in adipose tissue plays an important role in the development of metabolic alterations. Augmented oxidative stress reflects an imbalance between the production of reactive oxygen species (ROS) and antioxidant mechanisms [28]. Enhanced adipose tissue ROS are observed in increased adiposity due to an excess of electrons available in the mitochondrial electron transport chain, coupled with low ATP demand from decreased physical activity favoring the production of superoxide species [28,29]. Furthermore, prolonged adipose tissue oxidative stress impairs the endocrine and homeostatic functions of white adipose tissue by further disrupting adipokine secretion, limiting cellular antioxidant generation, and impairing mitochondrial function [28]. Thus, ROS generation is both inherent to the dysfunctional adipocyte and a byproduct of its dysfunction.

In summary, adipose tissue is a complex organ with critical endocrine and metabolic functions, in addition to its capacity for energy storage. In the setting of excess energy balance, adipose tissue expansion occurs through both hypertrophy and hyperplasia. In the visceral compartments, adipose tissue expansion occurs primarily through hypertrophy, which leads to the creation of larger, dysfunctional adipocytes with disordered signaling that serves as a catalyst for insulin resistance, metabolic dysfunction, and multisystem organ failure [22,23,24]. The mechanisms for this dysfunction include abnormal adipokine ratios, disordered communication with the surrounding extracellular matrix, preference for the creation of pro-inflammatory immune cells, and generation of reactive oxygen species [28,29]. Remaining to be understood are the specifics of how dysfunctional adipose tissue contributes to multisystem organ dysfunction, as observed in chronic lung diseases such as ILD.

## 3. Adiposity and ILD in Human Studies

Obesity as measured by BMI has been identified as an independent risk factor in the development of multiple subtypes of ILD [30,31,32]. However, this relationship is likely complex. Paradoxically, cohort studies examining the relationship between obesity and ILD mortality have shown a possible protective effect of obesity, with increased survival observed in patients with elevated body mass index (BMI) and decreased survival in patients with overall weight loss [33,34]. However, BMI is unlikely to be an accurate surrogate for body adiposity, as the volume and distribution of adipose tissue in various body compartments likely mediate the association among body weight, lung injury, and mortality in ILD. For example, loss of muscle mass from prolonged immobilization can lead to numerical weight loss and lower BMI but does not necessarily reflect changes in adipose tissue mass. This potential disconnection between changes in visceral adipose tissue and BMI could be the underlying reason for the paradoxical association between weight loss and increased overall mortality in the ILD cohort. This point requires future studies with clearly defined methods for measuring adiposity as an important consideration for determining associations with ILD incidence and progression.

### 3.1. Quantifying Adiposity

Measuring adipose tissue volume and the presence of dysfunctional adiposity is challenging. Popular metrics such as BMI fail to accurately reflect adipose tissue volume and the presence of dysfunctional adiposity. Anthropometric proxies for central obesity, such as waist to height ratio, which likely reflect dysfunctional adipocytes from visceral fat are usually associated with and are slightly better predictors of increased levels of health risk factors than BMI [35,36,37,38,39]. However, measuring visceral adiposity in the abdominal and thoracic cavities is difficult and rarely performed. Thus, studying the relationship between ILD and visceral adipose volume has been uncommon.

Highlighting the rarity of measuring visceral adiposity in ILD, there is only one study that measured pericardial and abdominal adiposity. This study was performed in patients with interstitial lung abnormalities (ILAs). ILAs are incidental findings observed on chest CT that are thought to be a precursor to ILD [40]. In an ILA cohort, investigators observed increased chest CT abnormalities and decreased forced vital capacity (FVC) in patients with increasing adipose tissue volume [41]. This observed effect was strongest for pericardial adipose tissue volume, which has a priori no direct impact on lung mechanics. This outcome suggests that the mechanism of impaired FVC may be related to the effect of visceral adipose tissue on lung inflammation [41]. Consistent with this hypothesis, cytokine profiles differed based on adipose distribution with increased levels of IL-6 observed in patients with higher volumes of pericardial and abdominal adiposity [41]. Similarly, leptin levels were increased in individuals with increasing abdominal adipose tissue volume [41]. Overall, this finding highlights the importance of measuring adiposity and its potential impacts on the induction of systemic inflammation, which might drive ILD pathogenesis.

### 3.2. Adiposity Reduction and Clinical Outcomes in ILD

Potential relationships exist between changes in adiposity and ILD outcomes. However, these relationships require clear measurement of adipose tissue volume, as well as definition of the mechanisms of weight loss. In a large observational cohort of patients with all types of ILD, weight loss of at least 1 kg (kg) was associated with increased one-year mortality risk across all BMI groups [42]. However, the study authors noted that observed weight loss trends did not consider the mechanism of weight loss, (e.g., intentional or unintentional) or the associated changes in adipose and muscle mass distribution [42]. Cachexia is a complication of a variety of chronic systemic illnesses, with increased inflammation leading to muscle wasting [43]. In IPF, decreased muscle mass in particular has been associated with higher mortality, leading to overall weight loss, but it does not reflect changes in visceral adipose tissue volume and distribution [44]. This fact emphasizes the importance of evaluating adiposity in these clinical study settings as a more accurate assessment of the impact of dysfunction adipocytes on ILD outcomes.

Although changes in BMI appear to be insufficient to reflect the inflammatory effects of adiposity, bariatric surgery reduces both BMI and total adipose tissue volume. In patients with ILD who undergo bariatric surgery, measures of lung function, including forced vital capacity (FVC) and diffusion capacity, are significantly improved [41,45]. In a small cohort of Japanese ILD patients who achieved weight loss with nutritional education and the adoption of healthier eating habits, there was also a significant improvement in pulmonary function testing [46]. Despite evidence that weight loss improves lung function and diffusion capacity, the mechanisms driving this improvement are unclear. It is difficult to separate improvements in underlying inflammation that could lead to improvements in lung function from improvements in lung mechanics due to decreased visceral thoracic and abdominal adiposity. Adiposity in the chest leads to diminished chest wall compliance, decreasing lung capacity [47]. In addition, the effect of thoracic adiposity on diffusion capacity is not as clear, with different studies showing decreased or unchanged diffusion capacity associated with increased adiposity [47]. Increased adiposity may lead to worsened V/Q mismatch through atelectasis or diminished lung volumes leading to decreased diffusion capacity [47,48]. Therefore, intentional weight loss in patients with ILD likely improves lung mechanics and diminishes overall inflammation, but how the latter occurs mechanistically remains unclear [41]. These findings support the concept that intentional weight loss in ILD patients to address the underlying metabolic derangements triggered by excess adiposity could potentially influence lung function and improve disease prognosis in ILD.

An alternate consideration for associations between adiposity and ILD is that increased ILD severity could increase dyspnea, with associated decreased calorie utilization that might drive increased adiposity. Currently, it is not clear whether individuals with ILD gain or lose adiposity on average with disease severity or over their disease course. It has been noted clinically that cachexia can be associated with significant ILD, similar to what has been observed with other chronic lung and cardiac diseases [42]. Additionally, data on ILAs associated with adiposity in different body compartments suggest that adiposity plays more of a causative, and not consequential, role in ILD, as patients would generally be minimally symptomatic from ILAs [41,43]. This research question remains largely open and requires clinical investigation.

### 3.3. Adiposity and Underlying Systemic Rheumatic Diseases

In addition to the complications of identifying appropriate objective surrogates for adipose tissue volume, the specific ILD subtype likely impacts the associations among morbidity, mortality, obesity, and adiposity. Select autoimmune causes of ILD relevant to this discussion have been described in the literature, but the association of adiposity and ILD has not been described for every rhematic disease associated with ILD.

In anti-synthetase syndrome-associated ILD, an increased burden of thoracic subcutaneous adipose tissue is associated with higher risk of relapse, as well as increased burden of fibrosis on imaging [49]. Furthermore, in this same study, patients with higher levels of thoracic adiposity were found to have higher levels of circulating KL-6, a marker of pulmonary epithelial injury [49,50].

In individuals with systemic sclerosis (SSc)-associated ILD (SSc-ILD), the adipocyte-derived adiponectin-related hormone, c1q/tumor necrosis factor-related protein 9 (CTRP9), may be a key mediator in SSc-ILD pathogenesis [51]. Although elevated CTRP9 levels were associated with decreased lung function, there was no correlation between CTRP9 levels and BMI in newly diagnosed SSc patients with ILD [51]. This finding again underscores the limitations of using BMI as a surrogate for adiposity, as BMI did not correlate with CTRP9 levels, and there were no measurements of visceral adipose tissue volume or distribution recorded in the study. In addition, resistin, another adipocyte-derived hormone observed to be elevated in Hispanic individuals with higher body adiposity, has been implicated in SSc-ILD pathogenesis and dermatomyositis-associated ILD [52]. SSc patients with ILD had higher resistin levels compared to healthy controls without ILD [53]. In dermatomyositis-associated ILD, patients with higher resistin levels also had decreased DLCO and more often had a rapidly progressive ILD [54]. Elevated resistin levels in adiposity may partially explain increased risk for a variety of different types of ILD.

Ultimately, this section highlights several gaps in clinical research knowledge in the area of ILD. In particular, there is a need to study how changes in body tissue composition, specifically changes in visceral adipose tissue mass, influence outcomes in ILD. Additionally, there have been no studies examining how changes in adipose tissue volume in ILD patients correlate with changes in adipokine signaling and pro-inflammatory immune system activity. There have also been no randomized, controlled clinical studies assessing the impacts of lifestyle, pharmacologic, or surgical interventions on adipose tissue volume and subsequent parameters related to ILD disease activity.

## 4. Inflammation and ILD

ILD refers to a group of chronic pulmonary disorders that principally affect the lung parenchyma [55]. ILD has two predominant but frequently overlapping phenotypes: inflammatory and fibrotic [56]. In inflammatory phenotypes, the accumulation of immune cells and other signaling molecules in the lung parenchyma leads to tissue damage. Inflamed lung tissue may promote the activation and proliferation of fibroblasts, as well as the inhibition of enzymes that break down extracellular matrix proteins [57]. The cumulative effect leads to scar formation, called “pulmonary fibrosis”. The mechanisms that drive inflammation and fibrosis in ILD are diverse and depend on the individual disease subtype. This process occurs via complex immune cell signaling that not only drives the initial development of disease but can also exacerbate established ILD. Much of this change occurs as a consequence of dysregulation of normal immune responses. As a high-adiposity state favors an overall inflammatory state, it is possible that these effects could worsen ILD incidence and progression, in cases both associated with and not associated with systemic autoimmunity, including IPF [58,59]. This interaction is highly complex, as adipose tissue includes immune cells that produce a variety of factors that may be systemically released. In addition, there are immune cells located in the lung tissue that directly impact ILD pathogenesis, particularly ILDs driven by inflammation or autoimmunity. How these immune cells interact and their impact on lung immunity largely remains to be clarified. Highlighting the effects of individual immune cells in ILD and the impact of adiposity on their function could therefore define potential mechanisms by which dysfunctional adipocytes promote the immune dysfunction that damages lung parenchyma.

### 4.1. Macrophages

Macrophages are abundant in adipose and lung tissue and are critical mediators of tissue immunity through diverse functions in maintaining homeostasis, initiating inflammation, and promoting inflammation resolution [60]. In adipose tissue, adipocyte size has a strong, positive correlation with the proportion of macrophages within adipose tissue [61]. Furthermore, macrophages in dysfunctional, enlarged adipocytes express a pro-inflammatory phenotype, which results in hypersecretion of pro-inflammatory cytokines, including TNF-α, IL-6, and IL-1, and decreased secretion of anti-inflammatory cytokines, such as IL-4, IL-10, and IL-13 [20,62,63]. In the adipose tissue, this macrophage phenotype promotes dysfunctional remodeling of the surrounding extracellular matrix (ECM). ECM remodeling is essential for normal adipose tissue expansion [64]. Excessive synthesis of inflexible, fibrillary components of the ECM and the inhibition of enzymes that degrade these tissues lead to adipose tissue fibrosis. This fibrosis furthers the pro-inflammatory, pro-oxidant environment that defines dysfunctional adipose tissue.

Macrophages also have important functions in the pathogenesis of ILD [65,66,67]. This fact is particularly true in autoimmune-related ILDs, in which infiltration of macrophages is a well-established phenomenon. In lung homeostasis, macrophages are located both in the airspace (i.e., alveolar macrophages) and in the interstitium (i.e., interstitial macrophages). Following injury or in the setting of inflammation, monocytes traffic into both the interstitium and the airspace and differentiate into macrophages. These monocyte-derived macrophages then can exhibit pro-inflammatory and pro-fibrotic functions [68,69,70,71,72]. These functions includes data suggesting that ILDs are associated with alterations in circulating monocytes. The overall presence of monocytes is associated with ILD outcomes, with increased monocyte counts associated with worse IPF outcomes [73]. In addition, in SSc-ILD, these monocytes exhibit mixed classical (i.e., expression of CD80, CD86, TLR2, and TLR4) and alternative (i.e., expression of CD206, CD204, and CD163) cell surface marker phenotypes, suggesting a heightened activation state [74]. Furthermore, the activation of monocytes appears to depend on a plasma-derived factor, as plasma from individuals with systemic sclerosis increased CCL2, IL-6, and TGF-β production and was able to activate fibroblasts [75]. How adiposity impacts the profile of these monocytes in ILD and their effects on monocyte-derived macrophage function remains to be clarified. Suggesting a potential interaction, in leptin-deficient (ob/ob) and diet-induced obesity models of obesity, there is evidence of increased circulating monocytes, particularly the inflammatory monocyte subsets and lung tissue monocytes [76]. A similar blood monocytosis has also been observed in humans with obesity [77]. Future evaluations need to consider whether there is a connection between this monocytosis observed in obesity and ILD pathogenesis.

### 4.2. T and B Lymphocytes

The connection among lymphocytes, adipose tissue, and lung inflammation remains controversial and poorly elucidated in the literature [78]. T cells, particularly CD4+ T cells, are critical cells involved in initiating, perpetuating, and polarizing inflammatory responses [58]. In this regard, T cells are critical mediators of chronic inflammation, which is a key feature of ILDs, particularly those with an autoimmune component. Related to adiposity, T cells are sensitive to metabolic signaling hormones, which are altered in the setting of excess adiposity [58,79,80]. T cell overstimulation by nutrient- and energy-sensing pathways may lead to augmented responses against self-antigens, which could promote autoimmunity [79]. In addition, CD4+ T cell activation can stimulate the activation of fibroblasts, leading to excess collagen deposition and tissue scaring [78]. Despite this potential, the specific mechanisms linking dysfunctional adipocytes and lung inflammation to T cell activation remain theoretical and require additional research.

B cells are another key component of the lymphoid arm of the immune system and are generally defined by their function in generating antibodies from prior pathogenic exposures. In autoimmune disease, B cells lose self-tolerance and generate pathogenic autoantibodies, which can lead to systemic organ dysfunction. B cells have been identified in lung parenchymal tissue adjacent to regions of fibrosis in different forms of ILD, including IPF, rheumatoid arthritis-associated ILD, SSc-ILD, and hypersensitivity pneumonitis [81,82,83,84]. Beyond the lung, there is evidence of interplay between B cells and adipose tissue, whereby B cells regulate the metabolic effects of excess adiposity, and adipose tissue alters B cell composition and function [85]. For example, excess adiposity is associated with an increase in so-called double negative B cells, which lack CD27 expression [86,87]. These “double negative B cells” are also upregulated in autoimmune diseases, such as systemic lupus erythematosus [87]. Excess adiposity is also associated with reduced production of IL-10, an anti-inflammatory cytokine produced by regulatory B cells [86]. Pro-inflammatory B-cells are also upregulated by leptin, which is present in larger quantities in states of excess adiposity driving increased production of the inflammatory cytokines IL-6 and TNF-α [88]. This finding suggests that excess adiposity may promote the production of autoantibodies and pro-inflammatory cytokines via B cell-derived mechanisms. However, a direct connection among altered B cell composition and function, adiposity, and the development and progression of ILD remains to be clarified.

### 4.3. Eosinophils and Neutrophils

Like T and B cells and macrophages, the mechanisms linking eosinophils and neutrophils to adiposity, inflammation, and ILD are poorly understood. In individuals with increased visceral adipose tissue mass, a preference for pro-inflammatory cytokine expression attracts neutrophils to the adipose tissue, which can contribute to local inflammation and further amplify the inflammatory response [58]. Eosinophils, which are present in normal adipose tissue, can be activated by the adipokine leptin in the adipose tissue, which is present in increased concentrations in dysfunctional, enlarged adipocytes [89]. Theoretically, this leptin-mediated activation of eosinophils could contribute to the inflammatory pathways of ILD, although this specific effect requires experimental testing [90]. This finding could have important implications for ILD, as prominent eosinophilia is observed in bronchial washings in multiple subtypes of ILD [90].

In conclusion, there is limited evidence to describe a direct cellular link between states of excess and dysfunctional adiposity and lung inflammation. However, many of the immune cells implicated in the pathogenesis of ILD are altered by states of excess adiposity, and it is reasonable to believe that dysfunctional adipose tissue plays some role in the development and progression of ILD. This association also represents a potential fertile area of future investigation to define potential mechanisms of crosstalk between dysfunctional adipose tissue and ILD.

## 5. Potential Mechanistic Pathways Linking Adiposity with ILD

Although sparse, presently available data from animal obesity models and in-vitro studies suggest potential mechanisms that link dysfunctional adipose tissue and the pathogenesis and prognosis of ILD. An overall schematic of proposed pathways and mediators of ILD linked to increased adiposity is shown in Figure 1.

### 5.1. Adiponectin, Leptin, and Resistin and ILD Development

Deficiency of adiponectin, an adipokine associated with metabolic regulation, tissue remodeling, and vascular homeostasis, has been implicated in the pathogenesis of SSc-ILD [91]. Adiponectin levels fall into states of excess adiposity [25]. Adiponectin-deficient mice experience spontaneous activation of alveolar macrophages with increased TNF-α and matrix metalloproteinase 12 (MMP-12) release, leading to the distortion of the architecture of distal lung airspaces [91,92].

While adiponectin levels fall into a state of excess adiposity, leptin increases [91]. Leptin stimulates proinflammatory cytokine release, as well as the activation of monocytes, dendritic cells, and macrophages [91]. Leptin increases transcription of profibrotic genes through an interaction with TGFβ, which is found in higher levels in bronchial fluid in correlation with higher leptin levels [93]. Mice with defective leptin receptor signaling are resistant to bleomycin-induced lung fibrosis, also suggesting that leptin signaling could regulate the pathogenesis of ILDs [93].

Resistin is another adipokine that has been linked to the development of systemic sclerosis-associated ILD [52]. In mice, resistin is mainly produced in adipocytes, although in humans it is also produced by peripheral mononuclear cells, macrophages, and myeloid cells [54]. Resistin promotes the production of several inflammatory cytokines, including IL-6, IL-1β, TNF-α, and IL-12 [53,54]. Various damage signals that lead to a T helper 2 (Th2) response may increase resistin expression, with resistin creating a positive feedback loop, leading to further activation of damage-associated molecular patterns (DAMPs) [94]. This Th2 response leads to lung inflammation and fibrosis [94]. Resistin is not found in the lungs of healthy humans, but increased lung expression has been found in patients with systemic sclerosis and asthma [94]. In bleomycin-induced mouse IPF models, resistin DNA expression was strongly associated with the degree of fibrosis [94]. Resistin has also been shown to reduce myofibroblast apoptosis, and knockout/overexpressing mice became resistant/more susceptible to pulmonary fibrosis in a bleomycin-induced model [94]. Resistin does appear to have a significant role in the development of ILD. However, while resistin is recognized as an adipokine, its role in adiposity-mediated ILD is less clear at this time, given that it is found in other tissues, in addition to adipocytes in humans.

### 5.2. Adiposity-Induced Vitamin D Deficiency

The impact of dysfunctional adipose tissue on vitamin D receptors in mouse models and in-vitro studies also offers a possible mechanism linking excess adiposity to ILD development. Upon binding to its receptor, vitamin D induces an array of downstream signaling effects that are important to maintaining healthy lung tissue [95]. These effects include the downregulation of the profibrotic cytokine TGFβ by inhibition of the phosphorylation of Smad-2/3 [95]. Vitamin D also directly suppresses transcription of renin and the downstream activation of the renin–angiotensin system [96]. Renin and angiotensin exert a variety of profibrotic effects on lung tissue, including the induction of mesenchymal growth factors and the promotion of motility in lung fibroblast cells [97].

Mice fed a high-fat, excess-calorie diet, compared to mice fed a standard diet, had significantly lower circulating vitamin D levels, as well as decreased mRNA expression of the vitamin D receptor in lung tissue [95]. These high-fat diet mice also had higher mRNA expression of TGFβ and higher levels of renin and angiotensin II in lung tissue, which were associated with increased peribronchial and perivascular fibrosis [95]. Hydroxyproline, a marker of collagen deposition and fibrosis, was also increased in the high-fat diet mice, along with other markers of collagen deposition [95]. Interestingly, these profibrotic trends in high-fat diet mice were ameliorated with vitamin D supplementation [95]. This animal model strongly suggests that adiposity-induced vitamin D deficiency has the potential to play a crucial role in the development and/or progression of ILD.

### 5.3. DOCK2 Signaling

Dedicator of cytokinesis 2 (DOCK2) is a positive regulator of the G protein Ras-related C3 botulinum toxin substrate (RAC) [98]. DOCK2 is mainly found in hematopoietic cells, and it regulates lymphocyte migration and activation, helper T cell differentiation, and interferon production [99]. DOCK2 may also be involved in energy metabolism. DOCK2-knockout mice have smaller body sizes, decreased insulin resistance, and increased brown and beige adipocyte accumulation [99]. In mice, DOCK2 expression increases with obesity. Conversely, DOCK2-knockout mice show diminished weight gain and reduced adipocyte hypertrophy [99]. Furthermore, DOCK2-knockout mice have diminished levels of the pro-inflammatory cytokines IL-6, IL-10, TNF-α, and IL-12, along with higher concentrations of anti-inflammatory adiponectin in adipose tissue [99].

In mouse obesity models, DOCK2 signaling is increased in lung tissue, concentrated in resident lung fibroblasts [100]. DOCK2-knockout mice have reduced lung macrophage infiltration and collagen production [100]. In human lung fibroblasts, DOCK2 knockout leads to decreased downstream inflammatory effects, even in the presence of TNF-α [100].

DOCK2 has also been implicated in the development of IPF. DOCK2 inhibition leads to diminished TGF-β-induced fibroblast-to-myofibroblast transformation in human fibroblast cells [101]. DOCK2 knockout also shows a protective effect in the setting of pro-fibrotic toxin exposure, as DOCK2 knockout ameliorates the damaging effects of bleomycin exposure in mouse models with decreased fibrosis [101].

Thus, DOCK2 expression is another example of a potential mechanism connecting adiposity and the development of ILD, as DOCK2 is found in higher concentrations in lung fibroblasts in states of excess adiposity, and its downregulation is shown to have protective effects against fibrotic damage.

### 5.4. STING Signaling

The stimulator of interferon genes, also known as STING, is a protein expressed in the endoplasmic reticulum [102]. STING is upregulated in the setting of lung damage, including in pulmonary fibrosis and acute infection [102]. Excess adiposity may enhance the expression of STING, resulting in increased downstream TNF-α, IL-6, and IFNβ production in lung tissues [102]. The mechanism for obesity-mediated increased STING appears to be a result of increased mitochondrial DNA leakage, as obese mouse models have increased mitochondrial DNA leakage into the cytosol through imbalanced levels of disulfide-bond A oxidoreductase-like protein [103]. Lung macrophages treated with palmitic acid, which is a fatty acid elevated in obesity, had increased mitochondrial DNA leakage into their cytosol, along with increased STING expression and downstream inflammatory cytokine levels [102]. STING knockout in this palmitic acid model did not fully normalize these inflammatory pathway changes, but it did ameliorate them [102]. Taken together, increased STING signaling due to impaired mitochondrial membrane integrity contributes to lung inflammation in the setting of increased adiposity. The exact molecular biology of this mitochondrial dysfunction is still under active research.

A summary of this section describing these mice and in-vitro studies linking adiposity and ILD can be found in Table 1. Further research is needed to further clarify these mechanisms in humans.

## 6. Conclusions

Dysfunctional adipose tissue promotes local and systemic inflammatory changes. Although there is limited information regarding the direct link between adipocyte dysfunction and ILD pathogenesis, there is evidence to suggest that dysfunctional adipocyte tissue has the capacity to generate an inflammatory and pro-fibrotic environment that could favor ILD development and progression. Adipose tissue has many effects on the immune system, including altering adipokine ratios, reducing vitamin D receptor activity, and directly influencing immune cell function. Adipose tissue may also alter the signaling pathways involved in the normal inflammatory response, promoting fibroblast activation and collagen deposition. The alterations of various inflammatory pathways in states of excess visceral adiposity share commonalities with the pathogenesis of ILD, and it can be postulated that excess visceral adiposity has implications for the development of ILD. Further study is needed to clarify the direct links between dysfunctional adipose tissue and the pathogenesis of ILD. These conclusions are limited in their applicability to all subtypes of ILD given the lack of information in the literature, with likely crucial differences in select subtypes. However, the pathways presented act as an overall summary of the link between adiposity and ILD development through increased lung inflammation. Additionally, there is a need for larger prospective studies or randomized, controlled studies examining how intentional weight loss, through lifestyle, pharmacologic, or surgical means, influences clinical outcomes. Furthermore, clarification of the direct impact of the altered immune functions of dysfunctional adipose tissue on the surrounding lung parenchyma could enable the development of targeted immune system modifications to improve outcomes in ILD, as well as provide useful clinical information for ILD across the spectrum.

## Figures and Tables

**Figure 1 cells-12-02206-f001:**
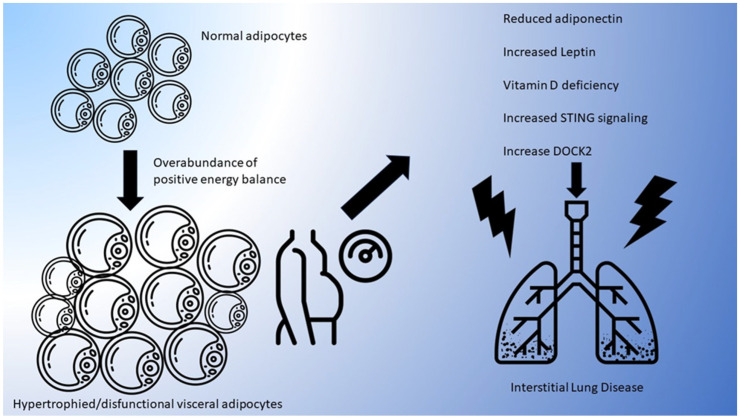
Hypothesized pathways linking adiposity and ILD. “Adipocyte” icons by dDara, from thenounproject.com CC BY 3.0, accessed on 15 June 2023. “Obesity” icon by Kukuh Wachyu Bias, from thenounproject.com CC BY 3.0, accessed on 15 June 2023. “Lungs” figure by Laymilk, from thenounproject.com CC BY 3.0, accessed on 15 June 2023. “Rays” icons by Aitor, from thenounproject.com CC BY 3.0, accessed on 15 June 2023.

**Table 1 cells-12-02206-t001:** Proposed adiposity-related disruptions tied to ILD development. TNF-α = tumor necrosis factor alpha, MMP-12 = matrix metallopeptidase 12, IFNβ = interferon beta, Th2 = T helper type 2, TGFβ = transforming growth factor beta, IL-6 = interleukin 6.

Proposed Adiposity-Related Disruptions Tied to ILD Development	Adiposity Influence on Mediator	Mediator Pathogenicity for ILD	Downstream Effects of Adiposity on Pathway
**Adiponectin**	Levels fall	Protective	Increased TNF-α and MMP-12 from less adiponectin
**Leptin**	Levels rise	Pathogenic	Increased inflammatory cytokines and profibrotic gene expression from more leptin
**Resistin**	Unclear effect on levels	Pathogenic	Increased inflammatory cytokines + Th2-mediated lung inflammation from more resistin Decreased Myofibroblast apoptosis from more resistin
**Vitamin D**	Levels fall	Protective	Increased TGFβ and renin from less vitamin D
**STING pathway**	Levels rise	Pathogenic	Increased TNF-α, IL-6, and IFNβ from more STING signaling
**DOCK2**	Levels rise	Pathogenic	Increased TGFβ effects, myofibroblast transformation, and inflammatory cytokines from increased DOCK2

## Data Availability

No new data were created or analyzed in this study. Data sharing is not applicable to this article.

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
