# Peer review of "Linking Adiposity to Interstitial Lung Disease: The Role of the Dysfunctional Adipocyte and Inflammation"

_cells, 2023, doi:10.3390/cells12182206_

Round 1
Reviewer 1 Report
This review article provides an update on the role of adipocyte in ILFs focusing on lung inflammation. The manuscript was well written and organized. A few concerns should be addressed to convey a better understanding of the topic to the readers.
1. With regards to the topic of adipocyte and ILD, it is recommended to briefly mention some relevant and current references that reported a similar topic but with different focuses. The most common form of ILF is IPF, obesity and IPF has recently been discussed and should be mentioned (PMID: 35082682).
2. Although the authors focus on adipocyte dysfunction and ILF pathogenesis, it is recommended to briefly mention some novel emerging molecules that play an important role in mediating both adipose tissue dysfunction such as adipose tissue inflammation and ILD such as IPF. A brief discussion of this point is recommended. Examples of relevant references (PMID: 28716822; PMID: 34767813; PMID: 35584329).
3. There are two different types of adipocytes, white and brown adipocytes. The authors are recommended to introduce the location of these two different types of adipocytes in the body. While much focus is on the white adipocytes, it would be good to briefly introduce adipose tissue browning as a means of improving health. Examples of reference to consider (PMID: 24146030).
Author Response
Reviewer one response to feedback:
We appreciate you taking the time to read our paper and for your valuable feedback. We have made edits based on your recommendations. Please refer to our most recent draft for changes and a detailed response to your comments.
- With regards to the topic of adipocyte and ILD, it is recommended to briefly mention some relevant and current references that reported a similar topic but with different focuses. The most common form of ILF is IPF, obesity and IPF has recently been discussed and should be mentioned (PMID: 35082682).
This reference has been incorporated. Our paper would in general also apply to IPF, this review is meant to be overall applicable to ILD and adiposity. There was some particular focus on connective tissue disease associated with ILD as there was some literature for these disease states, and due to several of the authors being practicing rheumatologists with interest in this area. Much of the material from the suggested reference is already addressed in the paper. We have clarified that the paper is meant to represent a general summary of the association between adiposity and ILD, and acknowledged the limitations in that not every subtype of ILD has research describing the role of adiposity.
- Although the authors focus on adipocyte dysfunction and ILF pathogenesis, it is recommended to briefly mention some novel emerging molecules that play an important role in mediating both adipose tissue dysfunction such as adipose tissue inflammation and ILD such as IPF. A brief discussion of this point is recommended. Examples of relevant references (PMID: 28716822; PMID: 34767813; PMID: 35584329).
A section on DOCK2 has been added, including advised references. As above we hope it is more that this paper would also in general apply to IPF as well.
Figures have also been updated with DOCK2.
Here is the main section that was added.
“Dedicator of cytokinesis 2 (DOCK2) is a positive regulator of the G protein Ras-related C3 botulinum toxin substrate (RAC) [98]. DOCK2 is mainly found in hematopoietic cells, and regulates lymphocyte migration and activation, helper T cell differentiation, and interferon production [99]. DOCK2 may also be involved in energy metabolism. DOCK2 knockout mice have smaller body size, decreased insulin resistance and increased brown and beige adipocyte accumulation [99]. In mice, DOCK2 expression increases with obesity. Conversely knockout DOCK2 mice show diminished weight gain, and reduced adipocyte hypertrophy [99]. Furthermore, DOCK2 knockout mice have diminished levels of the pro-inflammatory cytokines IL-6, IL-10, TNF-α and IL-12, along with higher concentrations of the anti-inflammatory adiponectin in adipose tissue [99].
In mice obesity models DOCK2 signaling is increased in lung tissue, concentrated in resident lung fibroblasts [100]. DOCK2 knockout mice have reduced lung macro-phage infiltration and collagen production PMID 34767813. In human lung fibroblasts, DOCK2 knockout leads to decreased downstream inflammatory effects even in the presence of TNF-α [100].
DOCK2 has also been implicated in the development of IPF. DOCK2 inhibition leads to diminished TGF-β induced fibroblast to myofibroblast transformation in hu-man fibroblast cells [101]. DOCK2 knockout also shows a protective effect in the setting of pro-fibrotic toxin exposure, as DOCK2 knockout ameliorates the damaging effects of bleomycin exposure in mouse models [101].
Thus, DOCK2 expression is another example of a potential mechanism connecting adiposity and the development of ILD, as DOCK2 is found in higher concentrations in lung fibroblasts in states of excess adiposity and its downregulation is shown to have protective effects against fibrotic damage.”
- There are two different types of adipocytes, white and brown adipocytes. The authors are recommended to introduce the location of these two different types of adipocytes in the body. While much focus is on the white adipocytes, it would be good to briefly introduce adipose tissue browning as a means of improving health. Examples of reference to consider (PMID: 24146030).
We do acknowledge that we focused on white adipose tissue in our paper due to a substantial lack of knowledge on how brown adipose tissue may interact with ILD, with the literature almost completely being based around the metabolic and cell signal changes associated with white adipose tissue. We have added a brief section outlining adult locations of brown adipose tissue and incorporated your suggested reference. The following new addition addresses this.
“Brown adipose tissue is found only in limited locations and quantities in adults. Subcutaneous locations include under the clavicles, in the axillary region, in the inguinal fossa, the anterior abdominal wall and the neck [17]. Visceral areas include perivascular locations, around solid organs and the trachea, esophagus, greater omentum and transverse colon [17]. White adipose tissue can also be induced to form beige adipocytes from cold temperatures or ꞵ3 signaling [18]. These beige adipocytes have similar thermogenic qualities as brown adipose tissue [18]. Brown adipocytes, and by extension beige adipocytes are thought to be protective against obesity and its resulting metabolic derangements by inducing energy utilization for thermogenesis [18]. It is an active area of study whether transplantation of brown adipose tissue, or inducing beige adipose tissue may be helpful for treating obesity[18]. Given the pathophysiology of white adipose tissue is much more studied and well described, we will herein address white adipose tissue only unless otherwise clarified.”
Reviewer 2 Report
Summary: The aim of the review by Macklin et al. was to detail the conditions and mechanisms by which adipose tissue might contribute to inflammatory changes observed in ILD. The review includes a summary of adipose tissue biology and then segways into potential mechanisms by which excess adiposity might contribute to the pathogenesis of ILD. Potential links between dysfunctional adipose tissue and immune cells, increases in proinflammatory adipokines, a reduction in adiponectin, and Vitamin D deficiency and STING upregulation in ILD are discussed. Finally, Table 1 summarizes the potential contribution of adiposity-related disruptions tied to ILD. The review provides a clear and comprehensive description of the current literature and its relevance to the field and there are no recent reviews that cover this topic. The conclusions are clearly drawn and gaps in knowledge are identified.
Specific Comments/Criticisms:
1) An issue that is not addressed is the confounding effects of ILD on physical activity and the potential for reverse causality. Does adipose tissue accumulate in patients with ILD due to reduced physical activity? This issue should be discussed.
2) Table 1 is confusing. The authors should use another what to clearly show how the levels of each mediator are affected by excess adiposity. For example, Adiponectin is an anti-inflammatory adipokine that declines with increased adiposity. However, the arrow next to “Adiposisty” is “↑” which implies that adiponectin increases with adiposity. It does not. There is also an “↑ TNF-ɑ and MMP-12” under the summary of adiposity effect on pathway which is also opposite to the effect of adiponectin on these mediators (R. Summer et al. reference #89). While it is true that alveolar macrophages from adiponectin knockout mice express higher levels of TNF-ɑ and MMP-12, adding adiponectin to alveolar macrophages reduced these mediators. mediators. In fact, including arrows next to adiposity and levels is confusing.
Author Response
Reviewer two response to feedback:
We appreciate you taking the time to read our paper and for your valuable feedback. We have made edits based on your recommendations. Please refer to our most recent draft for changes and a detailed response to your comments.
1.An issue that is not addressed is the confounding effects of ILD on physical activity and the potential for reverse causality. Does adipose tissue accumulate in patients with ILD due to reduced physical activity? This issue should be discussed.
We have added a brief section on this. This is a potential confounder though there are some counterpoints that would argue against this. Please see below for excerpt.
“An alternate consideration for associations between adiposity and ILD is that increased ILD severity could increase dyspnea with associated decreased calorie utilization that might drive increased adiposity. Presently, it is not clear if individuals with ILD gain or lose adiposity on average with disease severity, or over their disease course. It has been noted clinically that cachexia can be associated with significant ILD, similar to what is observed with other chronic lung and cardiac diseases [42]. Additionally, data on ILAs associated with adiposity in different body compartments suggests adiposity plays more of a causative and not consequential role in ILD as patients would generally be minimally symptomatic from ILAs [41,43]. This largely remains an open research question that requires clinical investigation.”
The literature does not include natural history studies with average weight changes over time in an ILD population. Baseline weight in several studies is reported but this is generally not an outcome measure. There is some literature in IPF in antifibrotic trials. In these studies, the percent of patients who lost weight is tracked due to side effects from these drugs, and some data indicating worse outcomes with weight loss as discussed in our paper, however weight from the overall group is not tracked so we cannot assess whether if on average patients stay the same weight or have gained weight.
Examples of this literature below:
PMID: 33098554, PMID: 32010718, PMID: 29490307, PMID: 36894966
2.Table 1 is confusing. The authors should use another what to clearly show how the levels of each mediator are affected by excess adiposity. For example, Adiponectin is an anti-inflammatory adipokine that declines with increased adiposity. However, the arrow next to “Adiposisty” is “↑” which implies that adiponectin increases with adiposity. It does not. There is also an “↑ TNF-ɑ and MMP-12” under the summary of adiposity effect on pathway which is also opposite to the effect of adiponectin on these mediators (R. Summer et al. reference #89). While it is true that alveolar macrophages from adiponectin knockout mice express higher levels of TNF-ɑ and MMP-12, adding adiponectin to alveolar macrophages reduced these mediators. mediators. In fact, including arrows next to adiposity and levels is confusing.
We apologize for any confusion. The intention of this table is to show what occurs with increasing adiposity, so for example as adiposity increases (up arrow) levels of adiponectin falls (down arrow) with the effect of adiposity leading to increased TNF-ɑ and MMP-12 due to the lower adiponectin levels from adiposity. Edits have been made to clarify the correct intentions of this table. There is also an additional section due to reviewer 1 comments with a new section on DOCK2.
Reviewer 3 Report
The idea of the paper is very intriguing. I am a pulmonologist and have enjoyed the part describing adipose tissue and related hormones. Other parts may be improved
Authors describe ILD in section 1. They write . "ILDs include several different types reflecting distinct etiologies. A large group of ILDs result from autoimmune diseases." They do not describe the key-variant of ILD, i.e. IPF. Similarly, in the following section they only discuss of ILDs related to rheumatic diseases. If authors prefer to discuss only of connective diseases related ILDs, they should change the title. Otherwise, they should describe IPF and other ILDs not related to connective diseases. In addition, they should report studies linking IPF and adiposity, if available.
I think that section 4 has been treated in previous sections. You should resume the argument and shorten the work. By contrast, you should enlarge the part describing the link between ILD and adiposity disorders, if available. Again, I find section 5 vague and not very interesting. I do not appreciate the link between the paragraphs "adiposity-induced vitamin D deficiency and STING and ILD/adipokines". Please underline key-point of this relationship. As a conclusion, i do not understand if is the link between adiposity and ILD only a suggestion or something more? Please discuss this point based on available evidence
Author Response
Reviewer three response to feedback:
We appreciate you taking the time to read our paper and for your valuable feedback. We have made edits based on your recommendations. Please refer to our most recent draft for changes and a detailed response to your comments.
1.Authors describe ILD in section 1. They write . "ILDs include several different types reflecting distinct etiologies. A large group of ILDs result from autoimmune diseases." They do not describe the key-variant of ILD, i.e. IPF. Similarly, in the following section they only discuss of ILDs related to rheumatic diseases. If authors prefer to discuss only of connective diseases related ILDs, they should change the title. Otherwise, they should describe IPF and other ILDs not related to connective diseases. In addition, they should report studies linking IPF and adiposity, if available.
This review is relevant to both connective tissue disease associated with ILD and IPF. The pre-clinical section on proposed pathogenic mechanisms specifically includes bleomycin lung fibrosis models, which are models for IPF. During this editing process, we have also added a paragraph to section 5 that specifically discusses a signaling mechanism identified in the pathogenesis of IPF that is also upregulated in states of excess adiposity called DOCK2. Outside of the section specifically labeled “Adiposity and underlying rheumatic conditions” this paper is meant to address the tie between ILD and adiposity, including IPF with limitations in this discussion due to lack of data in the literature available for every ILD subtype. These limitations are discussed. Some clarifications have been made throughout the paper.
2. I think that section 4 has been treated in previous sections. You should resume the argument and shorten the work. By contrast, you should enlarge the part describing the link between ILD and adiposity disorders, if available. Again, I find section 5 vague and not very interesting. I do not appreciate the link between the paragraphs "adiposity-induced vitamin D deficiency and STING and ILD/adipokines". Please underline key-point of this relationship. As a conclusion, i do not understand if is the link between adiposity and ILD only a suggestion or something more? Please discuss this point based on available evidence.
We appreciate your feedback. This paper is meant to blend the pre-clinical and clinical studies available linking adiposity to the development of ILD. Section 4 is a needed section, as its specific purpose is to review continues the narrative. Section 3 informs readers that adiposity is linked to inflammation. Section 4 informs how inflammation may lead to the development of ILD from the perspective of particular cell types that are pathogenically linked to ILDs. Section 5 links dysfunctional adipose tissue to the development of ILD through specific mechanisms that are well described in the literature. These mechanisms include the STING signaling pathway, vitamin D pathways, as well as changes in adipokine profiles. The STING pathway, which is upregulated in IPF, is also found to be upregulated in preclinical models of excess adiposity. Lung macrophages bathed in the fatty acid palmitic acid demonstrate increased downstream STING signaling. Vitamin D signaling is also an example of an overlap between adipocyte mediated dysfunction and ILD pathogenesis as pre-clinical studies demonstrate reduced vitamin D mRNA receptor expression in the lung tissue of mice with excess adiposity.
Ultimately, the link between ILD pathogenesis and dysfunctional adiposity is strongly suggested, but also still minimally understood. Our goal in this paper is to highlight not only the pre-clinical and clinical studies in this area, but also describe the overlap between adipocyte mediated inflammatory changes and inflammatory changes seen in ILD, which underscores the importance and potential fruitfulness of further research in this area. The pathways outlined in section 5 could be potential therapeutic targets for improving outcomes in many types of ILD. Pharmacologic and lifestyle changes for weight management may also improve ILD outcomes.
Round 2
Reviewer 2 Report
This reviewer is satisfied with the revisions.